# The Relationship of Family Cohesion and Teacher Emotional Support with Adolescent Prosocial Behavior: The Chain-Mediating Role of Self-Compassion and Meaning in Life

**DOI:** 10.3390/bs15081126

**Published:** 2025-08-19

**Authors:** Peng Li, Xia Zhou, Jiali Jiang, Shuying Fu, Xuejun Bai, Wenbin Feng

**Affiliations:** 1Key Research Base of Humanities and Social Sciences of the Ministry of Education, Academy of Psychology and Behavior, Tianjin Normal University, Tianjin 300387, China; 2Faculty of Psychology, Tianjin Normal University, Tianjin 300074, China; 3Tianjin Key Laboratory of Student Mental Health and Intelligence Assessment, Tianjin 300387, China; 4Jiangsu Provincial University Key Lab of Child Cognitive Development and Mental Health, Yancheng Teachers University, Yancheng 224002, China

**Keywords:** family cohesion, teacher emotional support, self-compassion, meaning in life, prosocial behavior

## Abstract

A questionnaire survey was conducted with 1153 adolescents to examine how emotional support within family and school contexts relates to adolescents’ prosocial behavior. Results indicated that both family cohesion and teacher emotional support were positively and significantly associated with prosocial behavior. Further analysis revealed that adolescents’ meaning in life mediated these relationships and that self-compassion together with meaning in life served as a sequential mediating pathway. When the direct effects of family cohesion and teacher emotional support on prosocial behavior were compared, teacher emotional support exhibited a significantly stronger direct association. However, no significant differences emerged between the two sources of support concerning the sequential (chain-mediating) pathways. These findings extend current understanding of adolescent prosocial development and highlight the importance of collaborative efforts by families and schools to meet adolescents’ emotional needs and promote prosocial tendencies.

## 1. Introduction

Prosocial behavior is that which benefits others and society, and is exhibited by individuals in social interactions ([11]). It can help adolescents adapt to society, reduce violent tendencies, decrease the risk of mental illness, and promote physical and mental health development. Cultivating prosocial behavior in adolescents holds significant practical implications for family and school education. From the perspective of positive adolescent development, adolescents can adapt and grow. When the environment provides sufficient support, they can actively participate in and achieve positive social roles. The outstanding performance of positive social roles in behavior is prosocial behavior, which can be achieved through internal psychological resource development supported by external resources ([26]). The existing research has focused on single dimensions of external resource support or internal psychological resource development while neglecting the interaction between internal and external dimensions. Therefore, this study integrated the dual systems of family and school and explored adolescent prosocial behavior from the perspective of the relationship between emotional support and psychological resources.

### 1.1. Family Cohesion and Teacher Emotional Support in Relation to Prosocial Behavior

Family and school, primary sources of social support, constitute a microsystem that significantly influences adolescents’ socialization and psychological resource development ([3]). Parents and teachers serve as crucial role models in adolescents’ developmental processes. When adolescents perceive a positive emotional connection with these role models, their internal beliefs and values are more likely to evolve, particularly during times of personal growth challenges or confusion. Emotional support from both family and school environments can help adolescents build a sense of self-empowerment and foster an understanding of meaning in life, thus facilitating the development of prosocial behaviors ([12]; [48]; [55]).

At the family level, family cohesion plays an important role in the socialization process of adolescents, promoting the formation of their prosocial behavior. On the one hand, research has found that as an important indicator of the family emotional system, family cohesion can not only reduce children’s loneliness and improve their prosocial behavior level through good parent–child interaction ([7]) but also help children perceive parental warmth and stably predict the level of self-compassion and prosocial behavior in adolescents and adults ([4]; [18]). On the other hand, [39] ([39]) found that adolescents who reported lower family cohesion also reported more violent behavior and less prosocial behavior, which suggests that family cohesion plays a role in shaping adolescents’ positive social behavior. Additionally, family cohesion, together with school belonging, forms a protective barrier for adolescents to cope with bullying and enhances their prosocial behavior level through emotional care ([38]).

At the school level, [12] ([12]) meta-analysis demonstrated that teacher–student relationships have a stronger effect than peer relationships on adolescents’ coping with adversity, peer relationship enhancement, and reduction in behavioral problems. Accordingly, this study prioritized teacher–student relationships within the school support system. The teacher’s emotional support in teacher–student relationships can enhance adolescents’ sense of belonging to the school, thereby promoting prosocial behavior ([1]). Specifically, teachers can create a good classroom atmosphere through their social and emotional abilities, demonstrating and guiding prosocial behavior in the process of resolving classroom disputes and encouraging adolescents to learn cooperation ([21]; [36]). This process not only helps students cope with academic difficulties but also enhances students’ emotional connection to the school, increases their sense of belonging, and motivates prosocial behavior ([1]; [29]). In summary, emotional support within the family and school systems is associated with adolescents’ prosocial behavior. Therefore, research hypothesis H1 is proposed: family cohesion and teacher emotional support are related to adolescents’ prosocial behavior.

### 1.2. The Mediating Role of Self-Compassion

Self-compassion refers to the support one gives to oneself when experiencing failure, suffering, or hardship due to one’s shortcomings or external challenges ([33]). For adolescents, family and school are important platforms for acquiring social support and learning self-compassion. For example, [54] ([54]) found that when adolescents encounter setbacks and failures, if they feel supported by their family and school, they will reduce their self-criticism, gradually develop self-compassion, and invest more energy in realistic activities, resulting in more constructive behaviors. Specifically, [42] ([42]) research showed that the higher the family cohesion of adolescents, the higher their level of self-compassion. [45] ([45]) found that a good teacher–student relationship can help students improve their self-compassion through teachers’ guidance and care when facing academic difficulties. Both studies demonstrate the close relationship between family–school support and adolescents’ self-compassion. The reason family–school support plays a role may be that emotional identification and resource provision provide a basic environment for the development of adolescents’ self-compassion and prosocial behavior ([28]). Therefore, when adolescents perceive social support from their family and school, they will produce positive interpersonal interactions, which can strengthen their emotional regulation ability and construct a cognitive schema of “self-worth being treated kindly.” This helps them to adopt a more tolerant self-dialogue mode when facing setbacks, and this self-accepting psychological state essentially expands their psychological resource capacity, providing the necessary conditions for implementing prosocial behaviors that require cognitive emotional resources ([26]). Thus, research hypothesis H2 is proposed: in the relationship between family cohesion, teacher emotional support, and adolescents’ prosocial behavior, self-compassion plays a mediating role.

### 1.3. The Mediating Role of Meaning in Life

The meaning in life refers to an individual’s perception of their existence and the essence of things, as well as their awareness of the importance and meaning of life ([40]). Adolescence, a critical stage of self-identity formation, derives its meaning in life from both the individual’s interactive experiences with the world and the ecological nurturing environment of family and school systems ([2]). Specifically, emotional support from parents in the family system can help adolescents reduce emotional alienation and significantly enhance their sense of existential meaning ([53]; [44]), thereby inspiring their meaning in life and promoting prosocial behavior ([50]). In the school system, schools serve as vital socialization domains. Teachers provide emotional support that meets adolescents’ need for belonging, which not only lessens their loneliness but also enhances the meaning in their lives by fostering a significant framework ([58]). The integrated model under the ecological system theory framework indicates that family cohesion can increase prosocial behavior through the mediating path of meaning ([27]). A three-stage cross-lagged study further revealed that there was a bidirectional dynamic relationship between adolescents’ meaning in life and prosocial behavior, providing empirical evidence for intervention practices in the school system ([51]). Therefore, research hypothesis H3 is proposed: in the relationship between family cohesion, teacher emotional support, and adolescents’ prosocial behavior, meaning in life plays a mediating role.

### 1.4. The Chain-Mediating Role of Self-Compassion and Meaning in Life

Existing studies have shown that self-compassion has a significant predictive effect on meaning in life. Self-compassion, defined as treating oneself with kindness, recognizing one’s shared humanity, and maintaining a balanced awareness of negative experiences ([33]), helps individuals reduce self-criticism and regulate emotions more effectively, thereby creating favorable psychological conditions for constructing life meaning ([41]). Empirical findings have demonstrated that self-compassion facilitates the presence of meaning in life, which mediates its link with reduced boredom ([34]) and greater life satisfaction ([59]). Multistep mediation models have further identified self-compassion as an antecedent to meaning in life in promoting positive mental health outcomes ([56]). Moreover, intervention research indicates that enhancing self-compassion can lead to improvements in life meaning among adolescents experiencing life stressors ([20]). Based on this theoretical and empirical evidence, we hypothesized self-compassion as the first mediator preceding meaning in life in our chain-mediation model.

It is worth noting that researchers have pointed out that self-compassion and meaning in life are both significantly related to positive emotional experiences, belonging to positive psychological resources ([8]; [43]). According to the theory of positive emotional expansion, individuals tend to engage in more prosocial behaviors when experiencing positive emotions ([10]; [14]). Therefore, this psychological association can be extended to prosocial behavior ([6]). Specifically, when adolescents face setbacks and failures, emotional support from parents and teachers can help them see their value, regain confidence and support for themselves, and reestablish their thinking about the meaning in life from the experience of setbacks and failures, producing more constructive behaviors ([23]; [56]). From empathy for others to self-compassion, the meaning in life is involved in this process, which not only helps individuals overcome boredom and achieve higher life satisfaction but also projects outward the energy of inner fulfillment, producing more prosocial behaviors ([59]). These findings suggest that self-compassion and meaning in life constitute positive psychological resources that predict prosocial behavior.

Therefore, research hypothesis H4 is proposed: in the relationship between family cohesion and teacher emotional support and adolescent prosocial behavior, self-compassion and meaning in life play a chain-mediating role. In summary, a research hypothesis model is proposed, as shown in Figure 1.

## 2. Methods

### 2.1. Participants

Cluster random sampling was implemented across six secondary schools in Tianjin, encompassing grades 7–12. Following the acquisition of written informed consent from parents and students, 1230 questionnaires were distributed. After exclusion of invalid responses (incomplete or patterned data), 1153 valid responses were retained (valid response rate = 93.7%). Participants (*N* = 1153) ranged from 12 to 18 years (*M* = 14.27, *SD* = 1.77), with 527 (45.7%) identifying as male.

### 2.2. Measures

#### 2.2.1. Prosocial Behavior

We used the revised version of the Prosocial Tendencies Measure, originally developed by [5] ([5]) and later revised by [24] ([24]) for use with Chinese adolescents. It consists of 26 items measuring six dimensions of adolescent prosocial behavior: emotional, compliant, altruistic, anonymous, public, and emergency. Items were rated on a 5-point Likert-type scale (1 = very unlike myself to 5 = very like myself), with higher scores indicating greater prosocial tendencies. In this study, the scale demonstrated excellent internal consistency (Cronbach’s α = 0.94).

#### 2.2.2. Family Cohesion

We used the revised version of the Family Cohesion Scale, originally developed by [35] ([35]) and revised by [13] ([13]) for use with Chinese adolescents. It consists of 16 items. Responses were recorded on a 5-point Likert-type scale (e.g., 1 = never to 5 = always), with higher scores indicating stronger family cohesion. Internal consistency in the current sample was excellent (Cronbach’s α = 0.90).

#### 2.2.3. Teacher Emotional Support

We used a middle school student perceived teacher emotional support questionnaire ([15]) consisting of 18 items measuring four dimensions: understanding, caring, respect, and encouragement. Responses were recorded on a 5-point Likert-type scale (e.g., 1 = strongly disagree to 5 = strongly agree), with higher scores reflecting greater perceived teacher emotional support. Internal consistency in the current sample was excellent (Cronbach’s α = 0.96).

#### 2.2.4. Self-Compassion

We used the revised version of the Self-Compassion Scale, originally developed by [37] ([37]) and later revised by [16] ([16]) for use with Chinese adolescents. It consists of 12 items measuring three dimensions: mindfulness, common humanity, and self-kindness. Items were rated on a 5-point scale (e.g., 1 = never to 5 = always), with higher scores indicating higher levels of self-compassion. In this study, the scale demonstrated good internal consistency (Cronbach’s α = 0.83).

#### 2.2.5. Meaning in Life

We used the revised version of the Meaning in Life Questionnaire, originally developed by [40] ([40]) and later revised by [47] ([47]) for use with Chinese adolescents. It consists of 10 items measuring two dimensions: seeking meaning and having meaning. Items were rated on a 7-point Likert-type scale (e.g., 1 = strongly disagree to 5 = strongly agree), with higher scores indicating higher levels of meaning in life. In this study, the scale demonstrated good internal consistency (Cronbach’s α = 0.83).

### 2.3. Data Processing and Analysis

We used SPSS 26.0 for common method bias testing, descriptive statistics, and correlation analysis, and Mplus 8.4 for the chain-mediating effect test.

## 3. Results

### 3.1. Common Method Bias Test

To address potential common method bias, several procedural steps were implemented: some items were reverse-scored, and anonymity and confidentiality were emphasized during data collection. Statistically, Harman’s single-factor test was conducted. The first unrotated factor explained 20.53% of the variance, which was below the critical threshold of 40% ([52]), suggesting no significant common method bias in this study.

### 3.2. Descriptive Statistics and Correlation Analysis

Descriptive statistics and correlations of variables are shown in Table 1. The results show that prosocial behavior, family cohesion, teacher emotional support, self-compassion, and meaning in life were all significantly positively correlated.

### 3.3. Chain-Mediation Analysis for Total Prosocial Behavior Score

After controlling for gender and age, a chain-mediation model was tested, with family cohesion and teacher emotional support as independent variables, prosocial behavior as the dependent variable, and self-compassion and meaning in life as mediators. The model demonstrated adequate fit: χ^2^(12) = 67.87, *p* < 0.001, CFI = 0.95, TLI = 0.93, SRMR = 0.04, RMSEA = 0.06 (90% CI [0.05, 0.07]). Standardized path coefficients are presented in Figure 2. Family cohesion (*β =* 0.15, *p* < 0.001) and teacher emotional support (*β =* 0.34, *p* < 0.001) significantly positively predicted adolescents’ prosocial behavior.

To examine the direct and indirect effects of family cohesion and teacher emotional support on prosocial behavior, we conducted mediation analyses using bias-corrected bootstrapping with 5000 resamples (95% CI). The results are presented in Table 2.

In the predictive path of family cohesion and teacher emotional support for prosocial behavior, the mediating effect of self-compassion was not significant (95% CI [0.002, 0.04], *p* = 0.09; 95% CI [0.001, 0.04], *p* = 0.09), and the mediating effect of meaning in life was significant (95% CI [0.01, 0.03], *p* = 0.002; 95% CI [0.01, 0.04], *p* = 0.004). The chain-mediating effect of self-compassion and meaning in life between family cohesion and prosocial behavior was significant (95% CI [0.01, 0.02], *p* = 0.003), and the chain-mediating effect between teacher emotional support and prosocial behavior was also significant (95% CI [0.004, 0.01], *p* = 0.003), with a total chain-mediating effect value of 0.01, accounting for 4.57% and 2.04% of the total effect, respectively. Further comparison of the direct predictive effects of family cohesion and teacher emotional support on prosocial behavior showed significant differences (c1–c2 = −0.20, 95% CI [0.11, 0.28], *p* < 0.001), while the comparison of chain-mediating effects showed no significant differences (ind3–ind6 = 0.001, 95% CI [−0.004, 0.001], *p* = 0.495), indicating that the predictive effects of family cohesion and teacher emotional support on adolescent prosocial behavior differed in both direct paths and chain-mediating paths with self-compassion and meaning in life.

### 3.4. Chain-Mediation Analyses for Prosocial Behavior Subdimensions

To further examine whether the proposed mediation model held for each specific type of prosocial behavior, we tested the model separately for each subdimension of the Prosocial Tendencies Measure ([5]; revised by [24]). Model fit indices and sequential indirect effects for subdimensions are presented in Table 3.

Results showed that the chain-mediation effect via self-compassion and meaning in life was significant for all subdimensions of prosocial behavior except “altruistic.” Specifically, family cohesion and teacher emotional support positively predicted self-compassion, which in turn predicted meaning in life, and ultimately promoted prosocial behavior, including public, anonymous, compliant, emotional, and dire behavior. The indirect effect for these subdimensions was significant, as indicated by bootstrap confidence intervals that did not contain zero. The model fit indices for all models were very similar, with CFI > 0.9, TLI > 0.9, RMSEA < 0.08, and SRMR < 0.05, indicating that the hypothesized model provided a good representation of the data for each subdimension.

## 4. Discussion

### 4.1. Family Cohesion and Teacher Emotional Support Predict Adolescent Prosocial Behavior

Consistent with hypothesis H1, results demonstrated significant positive associations between both family cohesion and teacher emotional support and adolescents’ prosocial behavior. These findings align with prior research ([1]; [7]) and ecological systems theory ([3]), which posits that family and school microsystems critically influence adolescent development. Family cohesion, reflecting emotional bonds within the family system ([13]), enhances adolescents’ ability to cope with challenges through supportive parent–child relationships ([7]). Similarly, teacher emotional support fosters school connectedness ([15]), regulating peer relationships and promoting prosocial behavior ([1]; [12]). Together, these proximal social support systems significantly predict increased prosocial behavior in adolescents.

The comparative analysis revealed differential predictive effects: In the direct path, teacher emotional support demonstrated a significantly stronger association with prosocial behavior than family cohesion. However, no significant difference emerged between these effects in the chain-mediation pathway involving self-compassion and meaning in life. These findings extend prior research by demonstrating domain-specific patterns. While [57] ([57]) observed greater school than family influences on academic procrastination, our prosocial behavior results similarly highlight teachers’ salient role in direct socialization. The nonsignificant mediation-path difference suggests that self-compassion and meaning in life may function as common mechanisms, accounting for shared variance between family and school predictors.

### 4.2. The Mediating Role of Self-Compassion

Contrary to hypothesis H2, self-compassion neither mediated the relationships between family cohesion/teacher support and prosocial behavior nor directly predicted prosocial behavior. These null findings diverge from [6] ([6]) results, but align with evidence suggesting self-compassion primarily facilitates prosocial behavior through reducing empathic distress ([33]). The nonsignificant effects may reflect measurement timing and focus differences. As adolescents require interpersonal engagement to perceive familial and academic support ([31]), their other-oriented focus during positive interactions might attenuate self-compassion’s direct effects. This construct’s inward orientation ([33]) potentially demands additional cognitive–emotional resources for attention redistribution toward others. Notably, while self-compassion showed no direct effects, it exhibited an indirect association via meaning in life. This pattern mirrors [31] ([31]) longitudinal findings, where self-compassion’s initial predictive effects diminished over time, highlighting its context-dependent and mediated nature in adolescent prosocial development.

### 4.3. The Mediating Role of Meaning in Life

Supporting hypothesis H3, meaning in life emerged as a significant partial mediator between both family cohesion and teacher support and prosocial behavior. These bidirectional relationships corroborate previous findings ([46]; [50]), highlighting meaning in life’s dual function as both an outcome of social support and a precursor to prosocial behavior. The current results extend prior research by demonstrating that meaning in life: (a) mediates between family rituals and prosocial behavior ([50]), (b) buffers against trauma-related distress ([46]), and (c) reciprocally enhances through prosocial acts ([22]; [51]). This suggests a positive-feedback loop wherein familial and academic emotional support elevates meaning in life, which subsequently promotes prosocial behavior that further reinforces meaning perception.

### 4.4. The Chain-Mediated Effect of Self-Compassion and Meaning in Life

Supporting hypothesis H4, analyses revealed a significant chain-mediation pathway whereby family cohesion and teacher emotional support predicted increased prosocial behavior through sequential enhancements in self-compassion and meaning in life. This reveals the connection between external resource support and the development of internal psychological resources in promoting adolescent prosocial behavior, thereby supporting the theory of positive adolescent development ([26]). On one hand, in the dimension of external resource support, significant others, such as parents and teachers, provide not only experiential information for implementing prosocial behavior but also create a prosocial atmosphere and offer emotional support to adolescents. This fosters their psychological development and maturity, enhancing their motivation for prosocial behavior ([30]). On the other hand, in the realm of psychological resource development, self-compassion can assist adolescents in better coping with setbacks and promote emotional maturity ([19]; [33]), while meaning in life can help them form perspectives about themselves and the world, encouraging personality maturation ([32]). These positive psychological resources enable adolescents to perceive emotional support in their environment, foster gratitude, and cultivate thoughts of reciprocating to society, thereby enhancing prosocial behavior ([49]). In this study, adolescents who perceived higher family cohesion and teacher emotional support were able to enhance prosocial behavior by increasing self-compassion and meaning in life. Moreover, self-compassion training can elevate adolescents’ meaning in life ([20]), suggesting that parents and teachers can aid adolescents in practicing self-support and reflecting on the meaning of life during times of failure and setbacks, thus transforming crises into opportunities for growth.

### 4.5. Practical Significance and Implications for Adolescent Development

Grounded in positive adolescent development theory, ecological systems theory, and the broaden-and-build theory of positive emotions, this study bridges external emotional support systems and internal psychological resources to reveal the mechanisms underlying adolescents’ prosocial behavior. Beyond its theoretical contributions, the findings carry important implications for families, schools, and broader adolescent-oriented social systems.

At the environmental level, our results confirm that both family cohesion and teacher emotional support are key facilitators of adolescents’ psychological growth. This suggests the need for enhanced family–school collaboration. Parents can contribute by creating emotionally supportive home environments that encourage adolescents to feel understood and cared for, fostering empathy and a desire to reciprocate such kindness. Teachers in turn can integrate emotional support into daily instruction through classroom climate, communication styles, and supportive interactions, helping students feel connected and valued.

Moreover, the chain-mediation results underscore the importance of cultivating self-compassion and meaning in life. These inner psychological resources are not only protective but also promotive factors for prosocial development. Educators and caregivers can promote self-compassion through reflective practices, emotion coaching, and compassionate feedback. Likewise, meaning-in-life education (e.g., through life narrative exploration, service learning, or guided reflection) can help adolescents develop a deeper understanding of purpose and community connection, thereby motivating prosocial action.

Finally, given the increasing reliance on remote learning and shifting educational settings, it is more important than ever for schools and families to jointly support adolescents’ emotional needs. Prosocial behavior is not only a product of personality but also an outcome of sustained exposure to emotionally supportive relationships and value-oriented environments. By aligning efforts across home and school, stakeholders can foster the social–emotional competencies that allow adolescents to thrive as engaged and compassionate members of society.

### 4.6. Limitations and Future Work

Despite its valuable contributions, this study has several limitations. First, although we examined the influence of family and school systems on adolescents’ prosocial behavior, peer relationships—an important socializing agent—were not included ([25]). Future research could investigate their potential moderating or mediating roles. Second, the exclusive reliance on adolescents’ self-reporting, despite using multidimensional measures of prosocial behavior, introduces the risk of social desirability bias ([9]). We did not employ construct-specific social desirability controls (e.g., distinct measures for prosocial behavior, self-compassion, or meaning in life). Future studies could improve precision by incorporating such differentiated scales. In addition, our chain-mediation analyses showed that the hypothesized pathway via self-compassion and meaning in life was significant for most prosocial subdimensions, but not for the altruistic dimension, suggesting that altruism may be shaped by additional factors—such as empathy, moral values, or situational cues—not captured in this study. Third, the cross-sectional design limits causal inference. Although we hypothesized self-compassion to precede meaning in life, supplementary analyses testing the reverse order yielded similar fit indices and indirect effects, suggesting a potentially reciprocal relationship. We retained the original ordering to align with theoretical reasoning and prior evidence ([33]; [41]; [56]). Future longitudinal or experimental research is needed to clarify the temporal dynamics between these constructs and their effects on prosocial behavior. Finally, religious beliefs may significantly influence adolescents’ existential perspectives ([17]), and future studies could explore how these factors contribute to meaning in life and prosocial behavior across diverse cultural and religious contexts.

## 5. Conclusions

Empirical evidence confirms that family cohesion and teacher emotional support robustly predict adolescent prosocial behavior. Meaning in life functions as both an independent mediator between these social resources and prosocial outcomes and as part of a sequential mediation chain with self-compassion. Specifically, elevated family cohesion and teacher emotional support predict heightened self-compassion, which subsequently enhances existential purpose, ultimately increasing adolescents’ prosocial behavior.

## Figures and Tables

**Figure 1 behavsci-15-01126-f001:**
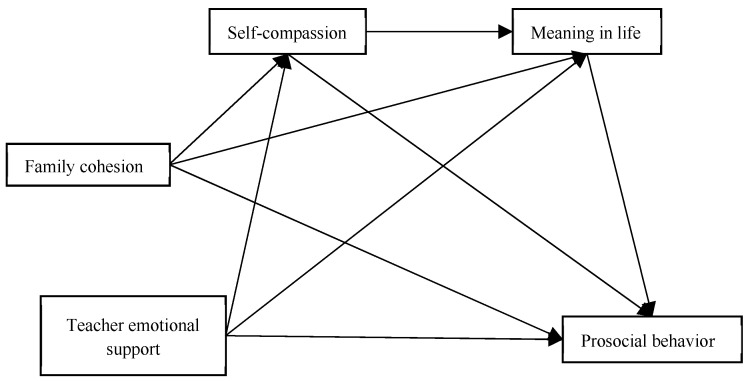
Hypothesis model.

**Figure 2 behavsci-15-01126-f002:**
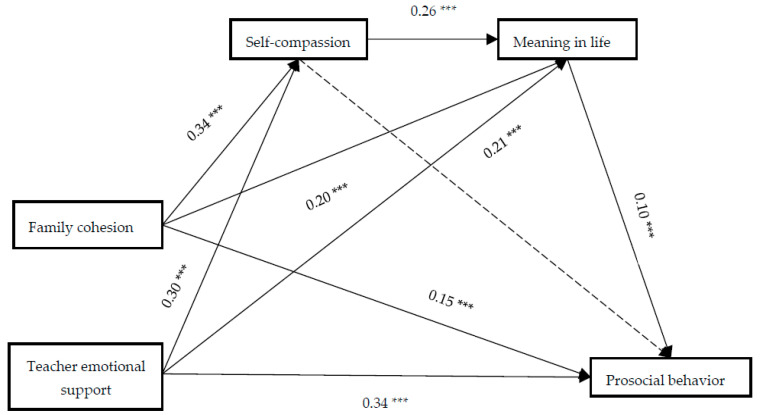
Chain-mediation model. *** *p* < 0.001.

**Table 1 behavsci-15-01126-t001:** Descriptive statistics and correlation analysis (*N* = 1153).

	*M* ± *SD*	1	2	3	4
1. Prosocial behavior	93.73 ± 17.38				
2. Family cohesion	56.89 ± 12.60	**0.37**			
3. Teacher emotional support	66.53 ± 14.19	**0.47**	**0.43**		
4. Self-compassion	40.28 ± 8.63	**0.32**	**0.47**	**0.44**	
5. Meaning in life	48.37 ± 10.79	**0.36**	**0.44**	**0.44**	**0.46**

Note: Bold formatting indicates statistically significant correlation coefficients.

**Table 2 behavsci-15-01126-t002:** Direct and indirect effects of family cohesion and teacher emotional support on prosocial behavior.

Independent Variable	Paths	Effect	SE	95% CI		Percentage of Total Effect (%)
				Lower	Upper	
X1	Total effect 1	0.20	0.03	0.14	0.24	
	c1: Direct (X1 → Y)	0.15	0.03	0.09	0.20	74.11
	ind1: X1 → M1 → Y	0.02	0.01	0.002	0.04	10.15
	ind2: X1 → M2 → Y	0.02	0.01	0.01	0.03	10.66
	ind3: X1 → M1 → M2 → Y	0.01	0.003	0.01	0.02	4.57
X2	Total effect 2	0.39	0.03	0.34	0.44	
	c2: Direct (X2 → Y)	0.34	0.03	0.29	0.40	87.76
	ind4: X2 → M1 → Y	0.02	0.01	0.001	0.04	4.59
	ind5: X2 → M2 → Y	0.02	0.01	0.01	0.04	5.61
	ind6: X2 → M1 → M2→ Y	0.01	0.003	0.004	0.01	2.04
Path Comparison	c1–c2	−0.20	0.05	0.11	0.28	
	ind3–ind6	0.001	0.002	−0.004	0.001	

Notes: X1 = family cohesion; X2 = teacher emotional support; M1 = self-compassion; M2 = meaning in life, Y = prosocial behavior (total score). Indirect effects were estimated using bias-corrected bootstrapping with 5000 resamples. CIs not containing zero indicate statistical significance.

**Table 3 behavsci-15-01126-t003:** Chain-mediation effects and model fit indices for prosocial behavior total score and subdimensions.

Dependent Variable	Independent Variable	Indirect Effect (X → M1 → M2 → Y)			χ^2^(df)	*p*	CFI	TLI	SRMR	RMSEA [90% CI]
		Effect	95% CI	Percentage of Total Effect (%)						
Total Score	X1 X2	0.009 0.008	0.005–0.015 0.004–0.013	4.57 2.04	67.874	<0.001	0.953	0.930	0.039	0.064 [0.050, 0.079]
Public	X1 X2	0.011 0.009	0.006–0.017 0.005–0.015	10 2.63	67.887	<0.001	0.946	0.919	0.039	0.064 [0.050, 0.079]
Anonymous	X1 X2	0.006 0.005	0.001–0.011 0.001–0.010	3.66 1.60	67.887	<0.001	0.947	0.921	0.039	0.064 [0.050, 0.079]
Altruistic	X1 X2	0.004 0.004	−0.001–0.010 −0.001–0.009	2.02 1.29	67.887	<0.001	0.949	0.923	0.039	0.064 [0.050, 0.079]
Compliant	X1 X2	0.006 0.005	0.001–0.011 0.001–0.010	3.66 1.55	67.887	<0.001	0.947	0.921	0.039	0.064 [0.050, 0.079]
Emotional	X1 X2	0.010 0.009	0.005–0.016 0.005–0.014	7.41 2.68	67.887	<0.001	0.947	0.920	0.039	0.064 [0.050, 0.079]
Dire	X1 X2	0.009 0.008	0.004–0.016 0.003–0.014	5.03 3.07	67.887	<0.001	0.945	0.917	0.039	0.064 [0.050, 0.079]

Notes: Model fit indices are reported only once per dependent variable. CIs not containing zero indicate statistical significance.

## Data Availability

The original data presented in the study are available via email request.

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
