# Peer review of "The Relationship of Family Cohesion and Teacher Emotional Support with Adolescent Prosocial Behavior: The Chain-Mediating Role of Self-Compassion and Meaning in Life"

_behavsci, 2025, doi:10.3390/bs15081126_

Round 1

Reviewer 1 Report

Comments and Suggestions for Authors

To the Authors,

Thank you for the opportunity to review your work on family cohesion, teacher emotional support, and prosocial behavior. Your study makes a nice contribution to the literature by integrating internal and external dimensions.

I was also impressed by the sample size of your data collection project and the novely of the mediating variables that you chose. Your study draws interesting conclusions about the role of teacher emotional support, another important socialization agent in adolescents’ lives, which has been not fully studied in the existing literature.

My suggestions to continue improving your paper are:

  • See if paragraphs can be divided into shorter ones in the background for clarity in reading
    • Maybe shorter paragraphs with subheadings for each variable
  • I would like to see more background research on why self-compassion and meaning in life were included as mediators in the order that they were in the chain mediation
    • Did the authors test a reverse causal model that could be included, in which you switch the order of self-compassion and meaning in life? I am not clear on why the authors expect self-compassion to come before meaning in life.
  • Even though the authors used a revised Prosocial Tendencies Measure, I feel it is important to give credit to the original authors of the Prosocial Tendencies Measure (Carlo & Randall, 2002).
    • It is typically not considered acceptable to create a composite variable of prosocial tendencies with all 6 dimensions of prosocial behavior. Public prosocial behavior is often considered a selfishly motivated form of prosocial behavior in the existing literature, and it would not be appropriate to lump all 6 forms into one composite.
    • In some previous literature, only dire, compliant, and emotional subscales have been used to create a composite variable of prosocial behavior.
  • In the Results section, I would like to see whether the Chi-square test of the model fit was significant.
  • Row names for tables should be left aligned for clarity

My main concern with this paper is the use of the Prosocial Tendencies Measure, as the purpose of the measure originally was to investigate the multidimensionality of prosocial behavior, and it was not intended to be used as a composite variable.

I look forward to reviewing the updates you make to your paper to continue to make it stronger.

Author Response

1. Summary

Thank you for your valuable feedback and constructive suggestions on our manuscript "The Relationship Between Family Cohesion and Teacher Emotional Support with Adolescent Prosocial Behavior: The Chain Mediating Role of Self-Compassion and Meaning in Life" (ID: behavsci-3754206). Your insights have been instrumental in enhancing the clarity and rigor of this work. We have carefully addressed all comments in the revised manuscript, with all modifications highlighted in red for your convenience. Detailed responses to each point are provided below.

2. Point-by-point response to Comments and Suggestions for Authors

General comments :

To the Authors,

Thank you for the opportunity to review your work on family cohesion, teacher emotional support, and prosocial behavior. Your study makes a nice contribution to the literature by integrating internal and external dimensions.

I was also impressed by the sample size of your data collection project and the novely of the mediating variables that you chose. Your study draws interesting conclusions about the role of teacher emotional support, another important socialization agent in adolescents’ lives, which has been not fully studied in the existing literature.

My suggestions to continue improving your paper are:

Comments 1: See if paragraphs can be divided into shorter ones in the background for clarity in reading. Maybe shorter paragraphs with subheadings for each variable.

Response 1: Thank you for your helpful suggestion. In response, we have revised the background section by dividing it into shorter paragraphs and adding subheadings for each key variable. This update appears in the Introduction.

1.1 Family Cohesion and Teacher Emotional Support in Relation to Prosocial Behavior

1.2 The Mediating Role of Self-Compassion

1.3 The Mediating Role of Meaning in Life

1.4 The Chain Mediating Role of Self-Compassion and Meaning in Life

Comments 2: I would like to see more background research on why self-compassion and meaning in life were included as mediators in the order that they were in the chain mediation. Did the authors test a reverse causal model that could be included, in which you switch the order of self-compassion and meaning in life? I am not clear on why the authors expect self-compassion to come before meaning in life.

Response 2: We appreciate your insightful comments. Our hypothesized sequence is grounded in both theoretical reasoning and empirical evidence. Therefore, we retained the hypothesized order of mediators in our model. We have also clarified this theoretical rationale in the Introduction (section 1.4, p 3, line 130-144) to strengthen the justification for our modeling decisions.

1.4 The Chain Mediating Role of Self-Compassion and Meaning in Life

Existing studies have shown that self-compassion has a significant predictive effect on the meaning in life. Self-compassion, defined as treating oneself with kindness, recognizing one’s shared humanity, and maintaining balanced awareness of negative experiences (Neff, 2023), helps individuals reduce self-criticism and regulate emotions more effectively, thereby creating favorable psychological conditions for constructing life meaning (Suh & Chong, 2022). Empirical findings have demonstrated that self-compassion facilitates the presence of meaning in life, which mediates its link with reduced boredom (O’Dea et al., 2022) and greater life satisfaction (Zipagan & Galvez, 2023). Multi-step mediation models have further identified self-compassion as an antecedent to meaning in life in promoting positive mental health outcomes (Yela et al., 2020). Moreover, intervention research indicates that enhancing self-compassion can lead to improvements in life meaning among adolescents experiencing life stressors (Isanejad et al., 2023). Based on this theoretical and empirical evidence, we hypothesized self-compassion as the first mediator preceding meaning in life in our chain mediation model.

In addition to our hypothesized model, we also tested the reverse ordering (meaning in life → self-compassion). This alternative model yielded significant results and similar fit indices. However, in the interest of maintaining a concise focus aligned with our theoretical framework, we chose not to present these findings in the main text. We acknowledge that exploring alternative causal orderings is a valuable avenue for future research and have added this note to the Discussion (section 4.6, p 10, line 395-402) to encourage further investigation.

“Third, the cross-sectional design limits causal inference. Although we hypothesized self-compassion to precede meaning in life, supplementary analyses testing the reverse order yielded similar fit indices and indirect effects, suggesting a potentially reciprocal relationship. We retained the original ordering to align with theoretical reasoning and prior evidence (Neff, 2023; Suh & Chong, 2022; Yela et al., 2020). Future longitudinal or experimental research is needed to clarify the temporal dynamics between these constructs and their effects on prosocial behavior. ”

Comments 3: Even though the authors used a revised Prosocial Tendencies Measure, I feel it is important to give credit to the original authors of the Prosocial Tendencies Measure (Carlo & Randall, 2002).

Response 3: Thank you for your important reminder. We fully agree that proper attribution to the original developers of the Prosocial Tendencies Measure (PTM) is essential. Accordingly, we have added a citation of Carlo and Randall (2002) in the Measures (Section 2.2, p 4-5, line 176-209) when introducing the revised version of the scale, to acknowledge their foundational contribution. In addition to citing the original authors of the PTM, we have revised all descriptions of revised scales in the Measures section to acknowledge both the original developers and the revisers, following academic citation standards.

“We used the revised version of the Prosocial Tendencies Measure, originally developed by Carlo and Randall (2002) and later revised by Kou et al. (2007) for use with Chinese adolescents.”

“We used the revised version of the Family Cohesion Scale, originally developed by Olson et al. (1982) and revised by Fei et al. (1991), for use with Chinese adolescents.”

“We used the revised version of the Self-Compassion Scale, originally developed by Raes et al. (2011) and later revised by Gong et al. (2014), for use with Chinese adolescents. ”

“We used the revised version of the Meaning in Life Questionnaire, originally developed by Steger et al. (2006) and later revised by Wang (2013), for use with Chinese adolescents.”

Comments 4: It is typically not considered acceptable to create a composite variable of prosocial tendencies with all 6 dimensions of prosocial behavior. Public prosocial behavior is often considered a selfishly motivated form of prosocial behavior in the existing literature, and it would not be appropriate to lump all 6 forms into one composite.

In some previous literature, only dire, compliant, and emotional subscales have been used to create a composite variable of prosocial behavior.

Response 4: We sincerely thank you for highlighting the importance of respecting the multidimensional nature of prosocial behavior in the use of the PTM. In response to this valuable feedback, we have revised the manuscript to conduct additional chain mediation analyses with each of the six PTM subdimensions as separate dependent variables (see Section 3.4 in the Results, p 7, line 260-277). This allowed us to examine whether the hypothesized mediation pathways via self-compassion and meaning in life hold consistently across different forms of prosocial behavior. The results showed that the chain mediation effects were significant for five subdimensions (public, anonymous, compliant, emotional, and dire), but not for the altruistic subdimension, underscoring the theoretical value of analyzing each form separately.

3.4. Chain Mediation Analyses for Prosocial Behavior Subdimensions

To further examine whether the proposed mediation model holds for each specific type of prosocial behavior, we tested the model separately for each subdimension of the Prosocial Tendencies Measure (Carlo & Randall, 2002; revised by Kou et al., 2007). The model fit indices and sequential indirect effects for each subdimension are presented in Table 4.

 Table 4. Chain Mediation Effects and Model Fit Indices for the Prosocial Behavior Total Score and Subdimensions

Results showed that the chain mediation effect via self-compassion and meaning in life was significant for all subdimensions of prosocial behavior except Altruistic. Specifically, family cohesion and teacher emotional support positively predicted self-compassion, which in turn predicted meaning in life, and ultimately promoted prosocial behavior, including public, anonymous, compliant, emotional, and dire behavior. The indirect effect for these subdimensions was significant, as indicated by bootstrap confidence intervals that did not contain zero. The model fit indices for all models were very similar, with CFI > 0.9, TLI > 0.9, RMSEA < 0.08, and SRMR < 0.05, indicating that the hypothesized model provided a good representation of the data for each subdimension.

At the same time, we have retained the composite score analysis. Because we consider the total score offers a parsimonious overview of the main model, while the subdimension analyses provide more nuanced insight. We have acknowledged the potential limitations of using a composite score in the Discussion (section 4.6, p 10, line 391-395) and highlighted the implications of the differential findings across subdimensions.

“In addition, our chain mediation analyses showed that the hypothesized pathway via self-compassion and meaning in life was significant for most prosocial subdimensions but not for the altruistic dimension, suggesting that altruism may be shaped by addi-tional factors—such as empathy, moral values, or situational cues—not captured in this study. ”

Comments 5: In the Results section, I would like to see whether the Chi-square test of the model fit was significant.

Response 5: Thank you for your suggestion. We have now included the Chi-square test of model fit in the Results section. The structural equation model demonstrated an acceptable fit to the data: χ2(12) = 67.87, p < 0.001, CFI = 0.95, TLI = 0.93, SRMR = 0.04, RMSEA = 0.06 (90% CI [0.05, 0.07]). Although the Chi-square test was statistically significant, the values of the other fit indices suggest that the model fits the data well. The relevant details have been added to Results (Section 3.3, p 6, line 228-234).

“After controlling for gender and age, a chain mediation model was tested with family cohesion and teacher emotional support as independent variables, prosocial behavior as the dependent variable, and self-compassion and meaning in life as mediators. The model demonstrated adequate fit: χ2(12) = 67.87, p < 0.001, CFI = 0.95, TLI = 0.93, SRMR = 0.04, RMSEA = 0.06 (90% CI [0.05, 0.07]). Standardized path coefficients are presented in Figure 2. Family cohesion (β = 0.15, p < 0.001) and teacher emotional support (β = 0.34, p < 0.001) significantly positively predicted adolescents' prosocial behavior.”

Comments 6: Row names for tables should be left aligned for clarity.

Response 6: Thank you for your helpful suggestion. We have adjusted the formatting of all tables in the manuscript. Specifically, the row labels have been left aligned to enhance clarity and improve the overall readability of the data presentation. This update appears in the Results (Section 3.3, p 6, line 241-244).

Table 2. Direct and Indirect Effects of Family Cohesion and Teacher Emotional Support on Prosocial Behavior.

Note: X1 = Family cohesion; X2 = Teacher emotional support; M1 = Self-compassion; M2 = Meaning in life, Y = Prosocial behavior of total score. Indirect effects were estimated using bias-corrected bootstrap with 5,000 resamples. CIs not containing zero indicate statistical significance. The same below.

Comments 7: My main concern with this paper is the use of the Prosocial Tendencies Measure, as the purpose of the measure originally was to investigate the multidimensionality of prosocial behavior, and it was not intended to be used as a composite variable. I look forward to reviewing the updates you make to your paper to continue to make it stronger.

Response 7: We appreciate your reiteration of the concern regarding the use of the PTM as a composite variable. As this point closely overlaps with Comment 5, we respectfully refer you to our detailed response to Comment 5. We thank you again for your thoughtful feedback.

Reviewer 2 Report

Comments and Suggestions for Authors

Dear authors,
The article you are submitting for publication is very interesting and topical.
A few suggestions:
- check how the bibliographic sources should be indicated in the text and at the end of this, following the journal requirements;
- it would be preferable for your article to include a subchapter regarding the implications of your results on society (young people, families, school, etc.) and what would be the ways in which prosocial behaviour can be encouraged among young people;
- I also consider it important to mention what the limits of your research and what suggestions/proposals you had for the development of this topic for the future.

Author Response

1. Summary

Thank you for your valuable feedback and constructive suggestions on our manuscript "The Relationship Between Family Cohesion and Teacher Emotional Support with Adolescent Prosocial Behavior: The Chain Mediating Role of Self-Compassion and Meaning in Life" (ID: behavsci-3754206). Your insights have been instrumental in enhancing the clarity and rigor of this work. We have carefully addressed all comments in the revised manuscript, with all modifications highlighted in red for your convenience. Detailed responses to each point are provided below.

2. Point-by-point response to Comments and Suggestions for Authors

General comments:

Dear authors,

The article you are submitting for publication is very interesting and topical.

A few suggestions:

Comments 1: - check how the bibliographic sources should be indicated in the text and at the end of this, following the journal requirements;

Response 1: Thank you for pointing this out. We have thoroughly reviewed the citation style throughout the manuscript and have revised both in-text references and the reference list to align with the formatting requirements of Behavioral Sciences. We appreciate your attention to detail.

Comments 2: - it would be preferable for your article to include a subchapter regarding the implications of your results on society (young people, families, school, etc.) and what would be the ways in which prosocial behaviour can be encouraged among young people;

Response 2: Thank you for this insightful and constructive suggestion. In response, we have revised the title of Section 4.5 (p 9, line 352-380) to better highlight the practical implications of our findings. We have also expanded this section to more explicitly discuss how emotional support from families and schools can foster adolescents’ prosocial behavior, and outlined concrete strategies through which parents and teachers can encourage self-compassion and meaning in life. These additions aim to clarify the societal relevance of our study and provide actionable insights for educational and developmental contexts.

4.5. Practical Significance and Implications for Youth Development

Grounded in positive youth development theory, ecological systems theory, and the broaden-and-build theory of positive emotions, this study bridges external emotional support systems and internal psychological resources to reveal the mechanisms underlying adolescents’ prosocial behavior. Beyond its theoretical contributions, the findings carry important implications for families, schools, and broader youth-oriented social systems.

At the environmental level, our results confirm that both family cohesion and teacher emotional support are key facilitators of adolescents’ psychological growth. This suggests the need for enhanced family–school collaboration. Parents can contribute by creating emotionally supportive home environments that encourage adolescents to feel understood and cared for, fostering empathy and a desire to reciprocate such kindness. Teachers, in turn, can integrate emotional support into daily instruction through classroom climate, communication styles, and supportive interactions, helping students feel connected and valued.

Moreover, the chain mediation results underscore the importance of cultivating self-compassion and meaning in life. These inner psychological resources are not only protective but also promotive factors for prosocial development. Educators and caregivers can promote self-compassion through reflective practices, emotion coaching, and compassionate feedback. Likewise, meaning in life education (e.g., through life narrative exploration, service learning, or guided reflection) can help adolescents develop a deeper understanding of purpose and community connection, thereby motivating prosocial action.

Finally, given the increasing reliance on remote learning and shifting educational settings, it is more important than ever for schools and families to jointly support adolescents’ emotional needs. Prosocial behavior is not only a product of personality but also an outcome of sustained exposure to emotionally supportive relationships and value-oriented environments. By aligning efforts across home and school, stakeholders can foster the social-emotional competencies that allow youth to thrive as engaged and compassionate members of society.

Comments 3: - I also consider it important to mention what the limits of your research and what suggestions/proposals you had for the development of this topic for the future.

Response 3: Thank you for your helpful suggestion. We agree that clearly outlining the study's limitations and proposing directions for future research is essential. In response, we have included a dedicated subsection within the discussion to address the study’s limitations and offer concrete suggestions for further exploration of this topic. This update appears in the Discussion (Section 4.6, p 10, line 383-405).

4.6. Limitations and Prospects of the Study

Despite its valuable contributions, this study has several limitations. First, although we examined the influence of family and school systems on adolescents’ prosocial behavior, peer relationships—an important socializing agent—were not included (Lee et al., 2017). Future research could investigate their potential moderating or mediating roles. Second, the exclusive reliance on adolescents’ self-reports, despite using multidimensional measures of prosocial behavior, introduces the risk of social desirability bias (Deutsch & Lamberti, 1986). We did not employ construct-specific social desirability controls (e.g., distinct measures for prosocial behavior, self-compassion, or meaning in life); future studies could improve precision by incorporating such differentiated scales. In addition, our chain mediation analyses showed that the hypothesized pathway via self-compassion and meaning in life was significant for most prosocial subdimensions but not for the altruistic dimension, suggesting that altruism may be shaped by additional factors—such as empathy, moral values, or situational cues—not captured in this study. Third, the cross-sectional design limits causal inference. Although we hypothesized self-compassion to precede meaning in life, supplementary analyses testing the reverse order yielded similar fit indices and indirect effects, suggesting a potentially reciprocal relationship. We retained the original ordering to align with theoretical reasoning and prior evidence (Neff, 2023; Suh & Chong, 2022; Yela et al., 2020). Future longitudinal or experimental research is needed to clarify the temporal dynamics between these constructs and their effects on prosocial behavior. Finally, religious beliefs may significantly influence adolescents’ existential perspectives (Hardy & Carlo, 2005); future studies could explore how these factors contribute to meaning in life and prosocial behavior across diverse cultural and religious contexts.

Reviewer 3 Report

Comments and Suggestions for Authors
  1. This is one of the best papers I've read in a long time, good job, authors!
  2. At line 101 would the word "require" be more accurate than "consume"?
  3. At line 309, I was once told that the Chinese character for crisis was a combination of the characters for danger and for opportunity, which seems to fit this sentence.
  4. One limitation of this paper, not mentioned, is that religion might play a role in one's meaning in life and that could be further studied in subsequent research.
  5. When discussing social desirability, keep in mind that you might need different measures for different constructs (i.e. marital social desirability for marital satisfaction, parental social desirability for parental satisfaction, individual social desirability for self-esteem, etc.).

Author Response

1. Summary

Thank you for your valuable feedback and constructive suggestions on our manuscript "The Relationship Between Family Cohesion and Teacher Emotional Support with Adolescent Prosocial Behavior: The Chain Mediating Role of Self-Compassion and Meaning in Life" (ID: behavsci-3754206). Your insights have been instrumental in enhancing the clarity and rigor of this work. We have carefully addressed all comments in the revised manuscript, with all modifications highlighted in red for your convenience. Detailed responses to each point are provided below.

2. Point-by-point response to Comments and Suggestions for Authors

Comments 1: This is one of the best papers I've read in a long time, good job, authors!

Response 1: We sincerely appreciate your kind and encouraging feedback. It is truly rewarding to know that our work resonated with you. Your recognition motivates us to continue conducting and sharing meaningful research.

Comments 2: At line 101 would the word "require" be more accurate than "consume"?

Response 2: Thank you for your thoughtful suggestion. We agree that "require" is a more accurate and appropriate word choice in this context. We have revised the sentence accordingly in the manuscript to improve clarity and precision. This update appears in the Introduction (Section 1.2, p 3, line 106).

“This helps them to adopt a more tolerant self-dialogue mode when facing setbacks, and this self-accepting psychological state essentially expands their psychological resource capacity, providing the necessary conditions for implementing prosocial behaviors that require cognitive emotional resources (Lerner et al., 2005).”

Comments 3: At line 309, I was once told that the Chinese character for crisis was a combination of the characters for danger and for opportunity, which seems to fit this sentence.

Response 3: Thank you for your insightful and culturally meaningful comment. This interpretation indeed aligns well with the message conveyed in our discussion. We appreciate your thoughtful engagement with our work.

Comments 4: One limitation of this paper, not mentioned, is that religion might play a role in one's meaning in life and that could be further studied in subsequent research.

Response 4: Thank you for your insightful suggestion. We agree that religion could indeed play a significant role in shaping one's sense of meaning in life, and we appreciate you highlighting this important aspect. While our current study did not address this variable, we acknowledge it as a limitation and will consider exploring its impact in future research. Therefore, we have updated the Discussion (Section 4.6, p 10, line 402-405) to reflect it.

“Finally, religious beliefs may significantly influence adolescents’ existential perspectives (Hardy & Carlo, 2005); future studies could explore how these factors contribute to meaning in life and prosocial behavior across diverse cultural and religious contexts.”

Comments 5: When discussing social desirability, keep in mind that you might need different measures for different constructs (i.e. marital social desirability for marital satisfaction, parental social desirability for parental satisfaction, individual social desirability for self-esteem, etc.).

Response 5: Thank you for this insightful suggestion. We agree that different constructs may require construct-specific measures of social desirability to more accurately assess and control for response bias. In our current study, we acknowledged the limitations of using self-report measures and the potential influence of general social desirability. However, we did not include differentiated social desirability scales tailored to specific constructs. We appreciate your comment and will consider incorporating such refined measurement approaches in future research to enhance the validity of our findings. Additionally, we have updated the manuscript in Discussion (Section 4.6, p 10, line 387-391).

Second, the exclusive reliance on adolescents’ self-reports, despite using multidimensional measures of prosocial behavior, introduces the risk of social desirability bias (Deutsch & Lamberti, 1986). We did not employ construct-specific social desirability controls (e.g., distinct measures for prosocial behavior, self-compassion, or meaning in life); future studies could improve precision by incorporating such differentiated scales.